# Association between gestational weight gain and severe adverse birth outcomes in Washington State, US: A population-based retrospective cohort study, 2004–2013

U. Vivian Ukah[1]*, Hamideh Bayrampour[2], Yasser Sabr[3], Neda Razaz[4], Wee-Shian Chan[5], Kenneth I. Lim[6], Sarka Lisonkova[6,7]

1 Department of Epidemiology, Biostatistics, and Occupational Health, McGill University, Montreal, Quebec, Canada, 2 Department of Family Practice, Faculty of Medicine, University of British Columbia, Vancouver, British Columbia, Canada, 3 Department of Obstetrics and Gynaecology, College of Medicine, King Saud University, Riyadh, Saudi Arabia, 4 Division of Clinical Epidemiology, Department of Medicine, Solna, Karolinska Institutet, Stockholm, Sweden, 5 Department of Medicine, University of British Columbia and BC Women's Hospital and Health Centre, Vancouver, British Columbia, Canada, 6 Department of Obstetrics and Gynaecology, University of British Columbia, BC Children's Hospital, and BC Women's Hospital and Health Centre, Vancouver, British Columbia, Canada, 7 School of Population and Public Health, University of British Columbia, Vancouver, British Columbia, Canada

* Vivian.Ukah@mail.mcgill.ca

**Data Availability Statement:** Analyses were based on administrative data collected and maintained by the Department of Health, State of Washington. We

## Abstract

### Background

Suboptimal weight gain during pregnancy is a potentially modifiable risk factor. We aimed to investigate the association between suboptimal gestational weight gain and severe adverse birth outcomes by pre-pregnancy body mass index (BMI) categories, including obesity class I to III.

### Methods and findings

We conducted a population-based study of pregnant women with singleton hospital births in Washington State, US, between 2004 and 2013. Optimal, low, and excess weight gain in each BMI category was calculated based on weight gain by gestational age as recommended by the American College of Obstetricians and Gynecologists and the Institute of Medicine. Primary composite outcomes were (1) maternal death and/or severe maternal morbidity (SMM) and (2) perinatal death and/or severe neonatal morbidity. Logistic regression was used to obtain adjusted odds ratios (AORs) and 95% confidence intervals. Overall, 722,839 women with information on pre-pregnancy BMI were included. Of these, 3.1% of women were underweight, 48.1% had normal pre-pregnancy BMI, 25.8% were overweight, and 23.0% were obese. Only 31.5% of women achieved optimal gestational weight gain. Women who had low weight gain were more likely to be African American and have Medicaid health insurance, while women with excess weight gain were more likely to be non-Hispanic white and younger than women with optimal weight gain in each pre-pregnancy BMI category. Compared with women who had optimal weight gain, those with low gestational

are not able to legally distribute the data specific to our analyses. The data is available from a third party, and a permission is required. NOTE: The Department of Health, State of Washington, charges a fee for reviewing the data request and preparing the requested dataset, including the documentation. We used this avenue to obtain the data (https://www.doh.wa.gov/DataandStatisticalReports/HealthcareinWashington/HospitalandPatientData/HospitalDischargeDataCHARS). A fee for data abstraction is common in many administrative databases (e.g., the Discharge abstract database housed by the Canadian Institutes for Health Information). We do not know about any other data source that would provide the same information free of charge.

**Funding:** This study was funded by Michael Smith Foundation for Health Research (website: https://www.msfhr.org/; Receiving author, SL; Grant no: 16888) and the Canadian Institutes of Health Research (website: http://www.cihr-irsc.gc.ca/e/193.html; Receiving author, SL; Grant no: F17-02161). The funders had no role in study design, data collection and analysis, decision to publish, or preparation of the manuscript.

**Competing interests:** The authors have declared that no competing interests exist.

**Abbreviations:** ACOG, American College of Obstetricians and Gynecologists; AOR, adjusted odds ratio; BMI, body mass index; ICD-9-CM, International Classification of Diseases–9th Revision, Clinical Modification; ICU, intensive care unit; IOM, Institute of Medicine; LGA, large for gestational age; SGA, small for gestational age; SMM, severe maternal morbidity.

weight gain had a higher rate of maternal death, 7.97 versus 2.63 per 100,000 ($p = 0.027$). In addition, low weight gain was associated with the composite adverse maternal outcome (death/SMM) in women with normal pre-pregnancy BMI and in overweight women (AOR 1.12, 95% CI 1.04–1.21, $p = 0.004$, and AOR 1.17, 95% CI 1.04–1.32, $p = 0.009$, respectively) compared to women in the same pre-pregnancy BMI category who had optimal weight gain. Similarly, excess gestational weight gain was associated with increased rates of death/SMM among women with normal pre-pregnancy BMI (AOR 1.20, 95% CI 1.12–1.28, $p < 0.001$) and obese women (AOR 1.12, 95% CI 1.01–1.23, $p = 0.019$). Low gestational weight gain was associated with perinatal death and severe neonatal morbidity regardless of pre-pregnancy BMI, including obesity classes I, II, and III, while excess weight gain was associated with severe neonatal morbidity only in women who were underweight or had normal BMI prior to pregnancy. Study limitations include the ascertainment of pre-pregnancy BMI using self-report, and lack of data availability for the most recent years.

## Conclusions

In this study, we found that most women do not achieve optimal weight gain during pregnancy. Low weight gain was associated with increased risk of severe adverse birth outcomes, and in particular with maternal death and perinatal death. Excess gestational weight gain was associated with severe adverse birth outcomes, except for women who were overweight prior to pregnancy. Weight gain recommendations for this group may need to be reassessed. It is important to counsel women during pregnancy about specific risks associated with both low and excess weight gain.

## Author summary

### Why was this study done?

- The Institute of Medicine (IOM) has guidelines for optimal weight gain during pregnancy for women who have normal body mass index (BMI) and those who are underweight, overweight, and obese prior to pregnancy. Suboptimal weight gain is associated with adverse health outcomes for the mother and baby.

- The prevalence of obesity is rising, yet knowledge gaps exist in gestational weight gain recommendations specifically for women with various degrees of obesity (class I to III).

- Limited information is available about severe maternal and neonatal morbidity in women with suboptimal weight gain in each pre-pregnancy BMI category, particularly in class I to III obesity.

### What did the researchers do and find?

- We conducted a study including all 722,839 women with singleton delivery in Washington State, US, between 2004 and 2013, to assess birth outcomes associated with suboptimal weight gain.

- Less than half of women achieved the recommended optimal gestational weight gain.

- Low gestational weight gain was associated with an increase in adverse perinatal outcomes, particularly with maternal death and stillbirth, and with an increase in severe maternal morbidity in normal weight and overweight women.

- Excess gestational weight gain was associated with an increase in adverse maternal and perinatal outcomes in women with pre-pregnancy BMI within or below the normal range; such association was not observed in women who were overweight prior to pregnancy, while severe maternal morbidity was elevated in obese women with excess weight gain.

- Among obese women, low weight gain was associated with perinatal death and/or severe neonatal morbidity in all obesity classes, while excess weight gain was associated with adverse maternal and perinatal outcomes in women who had class III obesity.

## What do these findings mean?

- The IOM weight gain recommendations for women who are overweight prior to pregnancy may need reassessment.

- Weight gain below the recommended range in women who are obese prior to pregnancy may increase the risk of perinatal death/severe neonatal morbidity.

- Low weight gain may potentially be used as a marker of elevated risk of stillbirth; however, further studies are needed to assess such utility.

## Introduction

Weight gained during pregnancy is a potentially modifiable risk factor for adverse pregnancy outcomes [1,2]. In 2009, the Institute of Medicine (IOM) published recommendations for optimal weight gain during pregnancy for women with singleton pregnancies, based on pre-pregnancy body mass index (BMI) [3]. These recommendations have been adopted by the American College of Obstetricians and Gynecologists (ACOG) guidelines for clinical practice to potentially reduce adverse outcomes during pregnancy for both mother and baby [1].

The association between low and excess gestational weight gain and perinatal outcomes has been studied with respect to small for gestational age (SGA), large for gestational age (LGA), macrosomia, neonatal seizures, low Apgar score, neonatal intensive care unit admission, and infant death [4–9]. And while the suboptimal weight gain is associated with preterm delivery [5,10–13], the majority of studies on gestational weight gain have included only term pregnancies [8,9,14,15], which largely underestimates the overall association between suboptimal weight gain and adverse pregnancy outcomes.

The effects of gestational weight gain on severe maternal morbidity (SMM), perinatal death, and severe neonatal morbidity have been understudied. Prior reports were relatively small and showed an elevated risk of cesarean delivery, preeclampsia, and blood transfusions among women who had excess weight gain during pregnancy [5,14,15]. One study reported an overall increased risk of SMM during hospital admission for delivery among women who had excess gestational weight gain; however, similarly to studies of perinatal outcomes, these

findings pertained only to term deliveries [16]. None of the prior studies included all serious adverse birth outcomes, including death and severe maternal and fetal/infant morbidity.

The prevalence of all types of maternal obesity continues to rise in developed countries [6,17–21]. The IOM/ACOG guidelines provide the same optimal weight gain recommendation for women with pre-pregnancy obesity regardless of the obesity class [1,3]. However, suboptimal weight gain likely differs with increasing BMI in these women [6,16], and this lack of specific recommendations by obesity class signifies an important knowledge gap.

This large population-based study investigated the associations between suboptimal gestational weight gain and maternal death and/or severe maternal morbidity, and perinatal death and/or neonatal morbidity, among women with various pre-pregnancy BMIs. In addition, we examined these severe adverse birth outcomes in women with pre-pregnancy class I, II, and III obesity.

## Methods

This study is reported according to the Strengthening the Reporting of Observational Studies in Epidemiology (STROBE) guideline (S1 STROBE Checklist). The study methodology and analysis plan (S1 Text) were developed prior to data manipulation and the actual analysis.

### Ethics

This study was exempted from ethics approval by the Washington State Department of Social and Health Services, US. According to the Washington State Institutional Review Board, a research study can be exempt from Washington State Institutional Review Board review if "the research does not involve obtaining data about subjects through interaction or intervention with individuals, and does not involve obtaining identifiable private information about subjects from confidential records." Our study was granted this exemption (DOH Exempt Request E-050515-H).

### Study population and data sources

Information on all singleton births between 20 and 45 weeks' gestation that occurred in Washington State hospitals from 1 January 2004 to 31 December 2013 was obtained from the Washington State Department of Health. Data sources included 2 linked population databases: BERD (Birth Event Record Database), including information from live birth and fetal death certificates, and CHARS (Comprehensive Hospital Abstract Reporting System), including birth hospitalization data. CHARS data provided information on delivery and newborn hospitalizations (up to 9 diagnostic and 9 procedure codes using the International Classification of Diseases–9th Revision, Clinical Modification [ICD-9-CM]), the type of health insurance coverage, intensive care unit (ICU) admission, and death during hospitalization. BERD included information on maternal characteristics such as pre-pregnancy BMI, age, race/ethnicity, education, marital status, parity, and obstetric history (previous infant death, preterm birth, or SGA birth in parous women); pregnancy characteristics such weight gain during pregnancy, smoking during pregnancy, assisted conception, gestational hypertension, and gestational diabetes; and birth characteristics such as year of birth, infant sex, gestational age at delivery, labour characteristics, mode of delivery, stillbirth, neonatal death, neonatal seizures, and congenital anomalies. Race/ethnicity was recorded in the birth certificate (BERD) as mother's self-identified race/ethnicity with the categories non-Hispanic white, black or African American, Native American or Alaska Native, and other (e.g., Asian: Indian, Chinese, Filipino, Japanese; also including "other" as an open-ended category); Hispanic origin was also self-reported and recorded as a separate category. All data were abstracted by trained abstractors using

standardized forms (S1 Form). Pre-pregnancy diabetes mellitus and pre-pregnancy hypertension were identified from both BERD and CHARS; the condition was classified as present if indicated in at least 1 dataset. The accuracy and completeness of the BERD (birth certificate) and CHARS (hospitalization) data was monitored by Washington State Department of Health through annual assessments and consistency checks [22]. Multiple births were excluded because twin or triplet sets could not be identified in the data source.

Self-reported pre-pregnancy weight and height was used to calculate BMI, classified according to the World Health Organization [23] into the following categories: normal BMI (18.5–24.9 kg/m$^2$), underweight (<18.5 kg/m$^2$), overweight (25.0–29.9 kg/m$^2$), and obesity class I (30.0–34.9 kg/m$^2$), class II (35.0–39.9 kg/m$^2$), and class III ($\geq$40 kg/m$^2$). BMI values were checked for consistency by the Department of Health: Maternal BMI values outside the expected range were flagged as potentially erroneous and rectified before the data were released. Gestational age at delivery was based on ultrasound dating; date of last menstrual period was used for women with missing ultrasound data.

## Weight gain during pregnancy

Pregnancy weight gain was defined as the woman's weight measured at childbirth minus her self-reported pre-pregnancy weight. Optimal weight gain was defined by IOM/ACOG guidelines [1]. For term pregnancy ($\geq$37 weeks), optimal weight gain was defined as the recommended range of total weight gain during pregnancy according to the pre-pregnancy BMI category: 28–40 lbs for underweight (<18.5 kg/m$^2$), 25–35 lbs for normal BMI (18.5–24.9 kg/m$^2$), 15–25 lbs for overweight (25.0–29.9 kg/m$^2$), and 11–20 lbs for obese ($\geq$30.0 kg/m$^2$). Women who delivered at term and had gestational weight gain lower or higher than the recommended weight gain were classified as having low or excess weight gain, respectively, i.e., suboptimal gestational weight gain. To account for gestational age at delivery among women who delivered preterm, optimal weight gain per week was calculated using the recommended weekly weight gain range according to the pre-pregnancy BMI category: 1.0–1.3 lbs for underweight, 0.8–1.0 lbs for normal BMI, 0.5–0.7 lbs for overweight, and 0.4–0.6 lbs for obese [1]. These recommendations pertain to the second trimester, assuming a weight gain range of 1.1–4.4 lbs in the first trimester; all calculations were performed consistently for each BMI category [1]. Comparisons were made between women with low and excess weight gain during pregnancy and those who achieved a recommended (optimal) weight gain.

## Outcomes

The primary outcomes were composite measures including severe adverse outcomes as follows: (1) maternal death (death during delivery hospitalization) and/or SMM and (2) perinatal death (including stillbirth or neonatal death within 28 days) and/or severe neonatal morbidity. SMM included maternal conditions that have a high case fatality rate or that lead to vital organ damage or serious long-term sequelae. A list of such conditions was developed by the Canadian Perinatal Surveillance System [24]; in addition, we included conditions recognized as SMM by the US Centers for Disease Control and Prevention [25–27], for instance, conditions requiring life-saving procedures such as mechanical ventilation or conversion of cardiac rhythm (see full list and definitions in S1 Table). Information on SMM was obtained from the CHARS dataset.

Severe neonatal morbidity was also identified from the CHARS database using ICD-9-CM diagnostic codes and included the following conditions: bronchopulmonary dysplasia, respiratory distress syndrome, retinopathy of prematurity, intraventricular hemorrhage grade 3 or more, periventricular leukomalacia, neonatal sepsis, necrotizing enterocolitis, and severe birth

trauma. The occurrence of any of these conditions or neonatal seizures (information obtained from birth certificates) composed severe neonatal morbidity.

## Statistical analyses

Logistic regression was used to obtain adjusted odds ratios (AORs) and 95% CIs expressing the association between pregnancy weight gain and the composite outcomes and their components stratified by pre-pregnancy BMI. AORs were adjusted for demographic and pre-pregnancy characteristics that are known to be associated with adverse maternal and perinatal outcomes. These included maternal age (<25 years, 25–34 years, or ≥35 years), maternal education (high school graduation or higher versus less than high school graduation), maternal marital status (single, widowed, or separated versus married or cohabitating), maternal race/ethnicity (Hispanic, African American, Native American, or other versus non-Hispanic white), parity (nulliparous or parity ≥ 4 versus parity 1–3), assisted conception (no versus yes), maternal smoking during pregnancy (no versus yes), type of health insurance (Medicaid versus private, self-pay, or other), year of birth, and fetal sex (female versus male). For neonatal outcomes, AORs were also adjusted for congenital anomalies.

## Supplementary analyses

In the supplementary analyses, we first examined associations between pregnancy weight gain and SMM components. We then assessed the associations between suboptimal weight gain and composite outcomes among obese women stratified by obesity class I, II, and III. We also assessed associations between weight gain and SGA and LGA, as secondary perinatal outcomes, to allow for comparisons with previous studies.

## Sensitivity analyses

We examined the associations between pregnancy weight gain and composite outcomes in a subset restricted to term pregnancies. The aim of this analysis was to facilitate comparisons with previous studies and to assess whether the results differ markedly from those for the total study population.

All analyses were performed using R statistical software (R version 3.5.1). Complete case analyses were performed (the number of records with any missing value was <3% of the study population). All p-values are reported as recommended by the American Statistical Association [28].

# Results

## Study population

Overall, 952,212 mothers gave birth (live birth or stillbirth) in Washington State between 1 January 2004 and 31 December 2013. We excluded births that occurred out of state, multiple births, births before 20 weeks' gestation, births to women aged <15 or >60 years (35,598 mothers, 3.7%), births that occurred out of hospital (24,716 mothers, 2.6%), and births that could not be matched with hospital records (64,609 mothers, 6.8%). From the remaining 827,289 births, 104,450 were excluded due to missing pre-pregnancy BMI or gestational weight gain (12.6%). The study population consisted of 722,839 women with a singleton pregnancy for whom information on pre-pregnancy weight and gestational weight gain was available. Of these, 3.1% were underweight, 48.1% had normal BMI, 25.8% were overweight BMI, and 23% were obese prior to pregnancy. Among the obese women, 56.6% had class I obesity, 26.7% had class II obesity, and 16.7% had class III obesity.

Overall, there were 143,509 (19.9%) women with low weight gain, 227,715 (31.5%) women with optimal weight gain, and 351,615 (48.6%) women with excess weight gain during pregnancy. Maternal demographic and clinical characteristics of women who had low, optimal, and excess weight gain differed by pre-pregnancy BMI.

### Low weight gain

Low weight gain was observed in 28.7% of women who were underweight, 21.1% of women with normal BMI, 14.1% of women who were overweight, and 22.6% of obese women (Fig 1). In general, women with low weight gain were more likely to be African American, more likely to have Medicaid health insurance, and less likely to be married than women with optimal weight gain in each pre-pregnancy BMI category. Except for underweight women, they were also more likely to be young, to have pre-pregnancy diabetes, and to smoke during pregnancy (Table 1). Overall, women with low weight gain had higher rates of spontaneous vaginal delivery and lower rates of prolonged labour. The rates of gestational diabetes were higher in these women, especially in those who were underweight and those with normal BMI prior to pregnancy (Table 2).

### Excess weight gain

Excess weight gain was observed in 26.3% of women who were underweight, 40.8% of women with normal BMI, 61.4% of women who were overweight, and 53.8% of obese women (Fig 1). In general, women with excess weight gain were more likely to be young, non-Hispanic white, and nulliparous than women who had optimal weight gain in each pre-pregnancy BMI

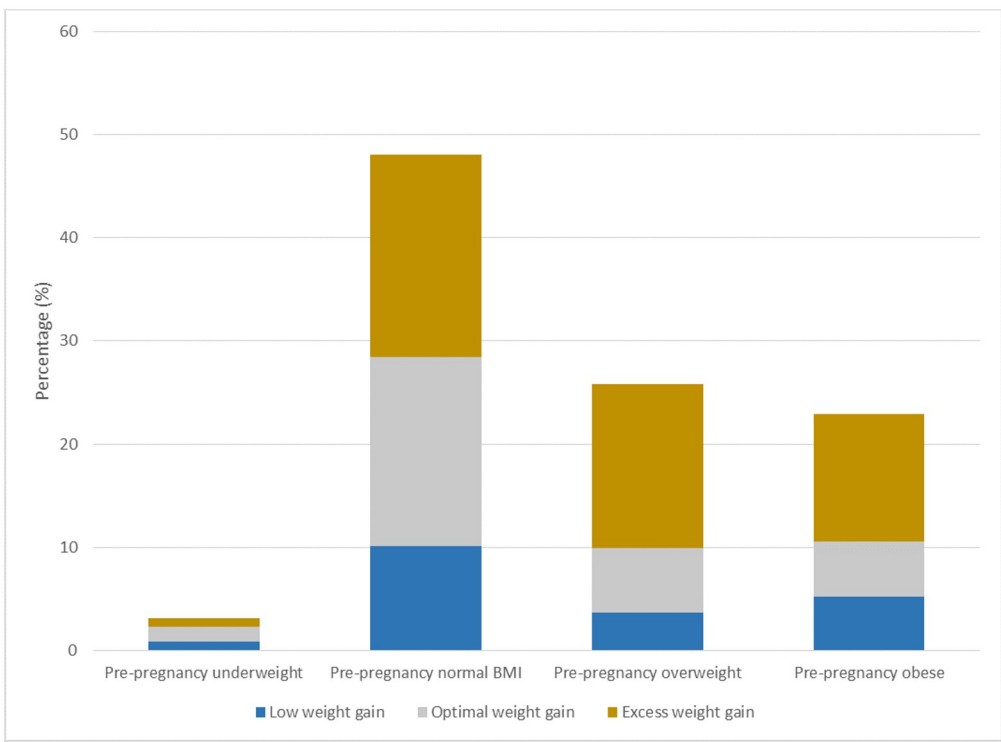

**Fig 1. Pre-pregnancy BMI category and gestational weight gain for singleton births, Washington State, 2004–2013.**

**Table 1. Demographic and pregnancy characteristics by pre-pregnancy BMI and gestational weight gain—Singleton births, Washington State, 2004–2013.**

| Characteristic | Pre-pregnancy underweight N = 22,714 (3.1%) | | | Pre-pregnancy normal BMI N = 347,555 (48.1%) | | | Pre-pregnancy overweight N = 186,662 (25.8%) | | | Pre-pregnancy obese N = 165,908 (23.0%) | | |
|---|---|---|---|---|---|---|---|---|---|---|---|---|
| | Optimal weight gain | Low weight gain | Excess weight gain | Optimal weight gain | Low weight gain | Excess weight gain | Optimal weight gain | Low weight gain | Excess weight gain | Optimal weight gain | Low weight gain | Excess weight gain |
| **Total population (percent of BMI category)** | 10,237 (45.1) | 6,510 (28.7) | 5,967 (26.3) | 132,572 (38.1) | 73,290 (21.1) | 141,693 (40.8) | 45,746 (24.5) | 26,284 (14.1) | 114,632 (61.4) | 39,160 (23.6) | 37,425 (22.6) | 89,323 (53.8) |
| **Maternal age, years** | | | | | | | | | | | | |
| 15–19 | 1,280 (12.5) | 810 (12.4) | 1,051 (17.6) | 9,428 (7.1) | 6,795 (9.3) | 13,949 (9.8) | 2,701 (5.9) | 1,929 (7.3) | 8,205 (7.2) | 1,551 (4.0) | 1,604 (4.3) | 5,392 (6.0) |
| 20–24 | 2,844 (27.8) | 1,786 (27.4) | 1,786 (33.7) | 25,838 (19.5) | 16,542 (22.6) | 33,370 (23.5) | 9,551 (20.9) | 5,907 (22.5) | 26,105 (22.8) | 8,490 (21.7) | 8,470 (22.6) | 22,052 (24.7) |
| 25–29 | 2,824 (27.6) | 1,719 (26.4) | 1,511 (25.3) | 37,758 (28.5) | 19,567 (26.7) | 40,760 (28.8) | 13,147 (28.74) | 7,369 (28.0) | 34,215 (29.8) | 12,214 (31.2) | 11,411 (30.5) | 27,442 (30.7) |
| 30–34 | 2,193 (21.4) | 1,386 (21.3) | 964 (16.2) | 37,232 (28.1) | 18,220 (24.9) | 34,510 (24.4) | 11,905 (26.0) | 6,383 (24.3) | 29,155 (25.4) | 10,133 (25.9) | 9,530 (25.5) | 21,625 (24.21) |
| 35–39 | 923 (9.0) | 642 (9.9) | 375 (6.3) | 18,285 (13.8) | 9,761 (13.3) | 15,851 (11.2) | 6,707 (14.7) | 3,650 (13.9) | 13,830 (12.1) | 5,384 (13.7) | 5,056 (13.51) | 10,374 (11.6) |
| ≥40 | 173 (1.7) | 167 (2.57) | 58 (1.0) | 4,031 (3.0) | 2,405 (3.3) | 3,253 (2.3) | 1,735 (3.8) | 1,046 (4.0) | 3,122 (2.7) | 1,388 (3.5) | 1,354 (3.6) | 2,438 (2.7) |
| **Race/ethnicity** | | | | | | | | | | | | |
| Non-Hispanic white | 6,657 (65.03) | 3,774 (58.0) | 4,351 (72.9) | 97,334 (73.4) | 46,579 (63.5) | 111,825 (78.9) | 30,095 (65.8) | 15,653 (59.5) | 88,435 (77.1) | 27,968 (71.4) | 26,212 (70.0) | 68,472 (76.7) |
| African American | 305 (3.0) | 291 (4.5) | 207 (3.5) | 3,303 (2.5) | 3,021 (4.1) | 3,778 (2.7) | 1,959 (4.3) | 1,647 (6.3) | 3,849 (3.4) | 1,969 (5.0) | 2,427 (6.5) | 3,755 (4.2) |
| Native American | 106 (1.0) | 74 (1.1) | 106 (1.8) | 1,460 (1.1) | 1,040 (1.4) | 2,270 (1.6) | 922 (2.0) | 604 (2.3) | 2,474 (2.2) | 1,206 (3.1) | 1,289 (3.4) | 2,748 (3.1) |
| Hispanic | 733 (7.2) | 661 (10.1) | 406 (6.8) | 11,759 (8.9) | 10,543 (14.4) | 9,860 (7.0) | 8,299 (18.1) | 5,288 (20.1) | 12,176 (10.6) | 5,857 (15.0) | 5,377 (14.4) | 9,712 (10.9) |
| Other | 2,405 (23.5) | 1,693 (26.0) | 864 (14.5) | 18,423 (13.9) | 11,937 (16.3) | 13,639 (9.6) | 4,368 (9.5) | 2,990 (11.4) | 7,438 (6.5) | 2,077 (5.3) | 2,018 (5.4) | 4,471 (5.0) |
| **Maternal education less than high school** | 220 (2.1) | 215 (3.3) | 140 (2.3) | 3,256 (2.46) | 3,543 (4.8) | 2,571 (1.8) | 2,924 (6.4) | 1,899 (7.2) | 3,534 (3.1) | 1,968 (5.0) | 1,583 (4.2) | 2,744 (3.1) |
| **Smoking** | 1,293 (12.6) | 759 (11.7) | 1,164 (19.5) | 9,187 (6.9) | 6,486 (8.8) | 14,780 (10.4) | 4,046 (8.8) | 2,941 (11.2) | 11,132 (9.7) | 4,364 (11.1) | 5,011 (13.4) | 10,794 (12.1) |
| **Married or cohabitating** | 6,776 (66.2) | 4,271 (65.6) | 3,119 (52.3) | 98,379 (74.2) | 48,806 (66.6) | 95,464 (67.4) | 30,618 (66.9) | 16,327 (62.1) | 76,480 (66.7) | 25,238 (64.45) | 23,331 (62.3) | 55,209 (61.8) |
| **Insurance** | | | | | | | | | | | | |
| Medicaid | 4,148 (40.5) | 2,928 (45.0) | 2,983 (50.0) | 41,078 (31.0) | 32,015 (43.7) | 47,936 (33.8) | 21,224 (46.4) | 14,078 (53.6) | 42,874 (37.4) | 19,074 (48.7) | 18,947 (50.6) | 39,406 (44.1) |
| Private | 5,269 (51.5) | 3,061 (47.0) | 2,445 (41.0) | 82,752 (62.4) | 36,124 (49.3) | 83,198 (58.7) | 21,420 (46.8) | 10,363 (39.4) | 63,299 (55.2) | 17,170 (43.8) | 15,490 (41.4) | 43,123 (48.3) |
| Self-pay | 114 (1.1) | 88 (1.3) | 75 (1.3) | 1,348 (1.0) | 863 (1.2) | 1,410 (1.0) | 387 (0.8) | 261 (1.0) | 980 (0.8) | 284 (0.7) | 303 (0.8) | 597 (0.7) |
| Other[a] | 647 (6.3) | 402 (6.2) | 445 (7.5) | 6,750 (5.1) | 3,957 (5.4) | 8,447 (6.0) | 2,532 (5.5) | 1,400 (5.3) | 6,966 (6.1) | 2,497 (6.4) | 2,440 (6.52) | 5,867 (6.6) |
| **Parity** | | | | | | | | | | | | |
| Nulliparous | 5,324 (52.0) | 3,060 (47.0) | 3,457 (57.9) | 57,911 (43.7) | 28,477 (38.9) | 73,305 (51.7) | 13,604 (29.7) | 8,165 (31.1) | 50,022 (43.6) | 10,680 (27.3) | 10,032 (26.8) | 35,644 (39.9) |
| 1–3 prior births | 4,687 (45.8) | 3,247 (49.9) | 2,308 (38.7) | 70,039 (52.8) | 41,276 (56.3) | 63,809 (45.0) | 28,911 (63.2) | 15,900 (60.5) | 59,369 (51.8) | 25,116 (64.1) | 23,745 (63.5) | 48,357 (54.1) |

(*Continued*)

**Table 1.** (Continued)

| Characteristic | Pre-pregnancy underweight N = 22,714 (3.1%) | | | Pre-pregnancy normal BMI N = 347,555 (48.1%) | | | Pre-pregnancy overweight N = 186,662 (25.8%) | | | Pre-pregnancy obese N = 165,908 (23.0%) | | |
|---|---|---|---|---|---|---|---|---|---|---|---|---|
| | Optimal weight gain | Low weight gain | Excess weight gain | Optimal weight gain | Low weight gain | Excess weight gain | Optimal weight gain | Low weight gain | Excess weight gain | Optimal weight gain | Low weight gain | Excess weight gain |
| ≥4 prior births | 174 (1.7) | 170 (2.6) | 142 (2.4) | 3,884 (2.9) | 3,106 (4.2) | 3,701 (2.6) | 3,005 (6.6) | 2,025 (7.7) | 4,571 (4.0) | 3,148 (8.0) | 3,376 (9.0) | 4,783 (5.3) |
| Assisted conception | 57 (0.6) | 49 (0.7) | 36 (0.6) | 1,451 (1.1) | 621 (0.9) | 1,412 (1.0) | 390 (0.8) | 169 (0.64) | 1,110 (1.0) | 371 (0.9) | 318 (0.9) | 838 (0.9) |
| Pre-pregnancy diabetes mellitus | 15 (0.2) | 10 (0.1) | 15 (0.3) | 364 (0.3) | 277 (0.4) | 587 (0.4) | 355 (0.8) | 238 (0.9) | 885 (0.8) | 765 (1.9) | 816 (2.2) | 1,750 (2.0) |
| Pre-pregnancy hypertension | 19 (0.2) | 23 (0.3) | 21 (0.3) | 529 (0.4) | 323 (0.4) | 761 (0.5) | 419 (0.9) | 214 (0.8) | 1,342 (1.2) | 1,276 (3.3) | 1,241 (3.3) | 2,876 (3.2) |
| Male fetal sex | 5,231 (51.1) | 3,150 (48.4) | 3,142 (52.7) | 67,192 (50.7) | 35,923 (49.0) | 75,215 (53.1) | 22,846 (49.9) | 13,031 (49.6) | 59,853 (52.2) | 19,810 (50.6) | 18,453 (49.3) | 46,758 (52.4) |
| Congenital anomalies[b] | 634 (6.2) | 422 (6.5) | 327 (5.5) | 7,411 (5.6) | 4,433 (6.0) | 7,930 (5.6) | 2,723 (5.9) | 1,715 (6.5) | 6,541 (5.7) | 2,444 (6.2) | 2,391 (6.4) | 5,466 (6.1) |

Data given as N (%). Underweight defined as BMI < 18.5 kg/m$^2$, normal weight as BMI 18.5–24.9 kg/m$^2$, overweight as BMI 25.0–29.9 kg/m$^2$, and obese class I–III as BMI ≥ 30.0 kg/m$^2$.

[a]Includes other government insurance, student insurance, Indian Health Service, and other programs.

[b]Includes the following conditions observed within first 24 hours after birth: anencephaly, meningomyelocele or spina bifida, cyanotic congenital heart disease, congenital diaphragmatic hernia, omphalocele, gastroschisis, limb reduction, cleft lip, cleft palate, Down syndrome, chromosomal disorders, and hypospadias.

category (Table 1). Except for underweight women, women with excess weight gain were more likely to have private health insurance. Among those with low and normal pre-pregnancy BMI, women who had excess weight gain were more likely to smoke during pregnancy and less likely to be married than women with optimal weight gain (Table 1). Women with excess weight gain were also more likely to have gestational hypertension, chorioamnionitis, induced labour, premature rupture of membranes, and cesarean delivery (Table 2).

## Maternal outcomes in women with low gestational weight gain

Overall, women with low gestational weight gain had a significantly higher rate of maternal death than women with optimal weight gain (7.97 versus 2.63 per 100,000, $p$ = 0.027). The rates of maternal death/SMM were 1.57%, 1.55%, 1.90%, and 1.75% among underweight, normal weight, overweight, and obese women, respectively (S2 Table). Among normal weight and overweight women, those with low gestational weight gain had higher adjusted odds of maternal death/SMM compared with those who had optimal gestational weight gain (AOR 1.12, 95% CI 1.04–1.21, $p$ = 0.004, and AOR 1.17; 95% CI 1.04–1.32, $p$ = 0.009, respectively; Table 3). Women who had normal pre-pregnancy BMI and low weight gain had elevated odds of cerebrovascular/central nervous system morbidity, while overweight women had increased odds of ICU admission (S4 Table).

## Maternal outcomes in women with excess gestational weight gain

The maternal death rate among women with excess weight gain was 3.98 per 100,000 (versus 2.63 per 100,000 in the optimal weight gain group, $p$ = 0.394). The rates of maternal death/SMM were 1.94%, 1.54%, 1.56%, and 1.81% among underweight, normal weight, overweight, and obese women, respectively (S2 Table). Adjusted for other covariates, among underweight, normal weight, and obese women, excess gestational weight gain was associated with elevated

**Table 2. Labour and delivery characteristics by pre-pregnancy BMI and gestational weight gain—Singleton births, Washington State, 2004–2013.**

| Characteristic | Pre-pregnancy underweight N = 22,714 (3.1%) | | | Pre-pregnancy normal BMI N = 347,555 (48.1%) | | | Pre-pregnancy overweight N = 186,662 (25.8%) | | | Pre-pregnancy obese N = 165,908 (23.0%) | | |
|---|---|---|---|---|---|---|---|---|---|---|---|---|
| | Optimal weight gain | Low weight gain | Excess weight gain | Optimal weight gain | Low weight gain | Excess weight gain | Optimal weight gain | Low weight gain | Excess weight gain | Optimal weight gain | Low weight gain | Excess weight gain |
| Total population (percent of BMI category) | 10,237 (45.1) | 6,510 (28.7) | 5,967 (26.3) | 132,572 (38.1) | 73,290 (21.1) | 141,693 (40.8) | 45,746 (24.5) | 26,284 (14.1) | 114,632 (61.4) | 39,160 (23.6) | 37,425 (22.6) | 89,323 (53.8) |
| **GA at delivery** | | | | | | | | | | | | |
| 20–33 weeks | 154 (1.5) | 184 (2.8) | 162 (2.7) | 1,265 (0.9) | 1,416 (1.9) | 2,761 (2.0) | 579 (1.3) | 734 (2.8) | 1,786 (1.6) | 663 (1.7) | 1,094 (2.9) | 1,634 (1.8) |
| 34–36 weeks | 559 (5.4) | 519 (8.0) | 437 (7.3) | 4,595 (3.5) | 3,612 (4.9) | 9,359 (6.6) | 1,701 (3.7) | 1,619 (6.2) | 6,077 (5.3) | 1,901 (4.9) | 2,475 (6.6) | 5,412 (6.1) |
| ≥37 weeks | 9,524 (93.0) | 5,807 (89.2) | 5,368 (90.0) | 126,712 (95.6) | 68,262 (93.1) | 129,573 (91.5) | 43,466 (95.0) | 23,931 (91.1) | 106,769 (93.1) | 36,596 (93.5) | 33,856 (90.5) | 82,277 (92.1) |
| **Vaginal delivery** | | | | | | | | | | | | |
| Spontaneous | 7,379 (72.1) | 4,794 (73.6) | 4,083 (68.4) | 92,888 (70.1) | 52,508 (71.6) | 93,472 (66.0) | 30,477 (66.62) | 18,115 (68.9) | 71,393 (62.3) | 22,888 (58.4) | 22,496 (60.1) | 48,399 (54.2) |
| VBAC | 106 (1.0) | 74 (1.1) | 56 (0.9) | 1,933 (1.5) | 1,175 (1.6) | 1,681 (1.2) | 905 (1.98) | 491 (1.9) | 1,649 (1.4) | 663 (1.7) | 636 (1.7) | 1,185 (1.3) |
| Forceps | 112 (1.1) | 68 (1.0) | 58 (1.0) | 1,147 (0.9) | 509 (0.7) | 1,330 (0.9) | 228 (0.5) | 136 (0.5) | 806 (0.7) | 163 (0.4) | 142 (0.4) | 455 (0.5) |
| Forceps VBAC | 2 (0.0) | 2 (0.0) | 0 (0.0) | 32 (0.0) | 21 (0.0) | 35 (0.0) | 19 (0.04) | 5 (0.02) | 28 (0.02) | 10 (0.0) | 15 (0.0) | 15 (0.0) |
| Vacuum | 719 (7.0) | 417 (7.0) | 468 (7.8) | 7,415 (5.6) | 3,715 (5.6) | 8,226 (5.8) | 1,766 (3.9) | 1,063 (3.9) | 5,070 (4.4) | 1,133 (2.8) | 1,144 (2.9) | 2,910 (3.3) |
| Vacuum VBAC | 19 (0.2) | 10 (0.1) | 12 (0.2) | 233 (0.2) | 133 (0.2) | 167 (0.1) | 100 (0.2) | 50 (0.2) | 171 (0.2) | 61 (0.2) | 51 (0.1) | 102 (0.1) |
| **C-section** | | | | | | | | | | | | |
| Primary with labour | 633 (6.2) | 345 (5.3) | 504 (8.4) | 9,287 (7.0) | 4,028 (5.5) | 14,271 (10.1) | 3,325 (7.3) | 1,539 (5.9) | 12,692 (11.1) | 3,657 (9.3) | 3,165 (8.5) | 12,149 (13.6) |
| Primary with no labour | 722 (7.0) | 440 (6.8) | 465 (7.8) | 9,608 (7.2) | 5,422 (7.4) | 12,368 (8.7) | 3,556 (7.8) | 2,219 (8.4) | 10,938 (9.5) | 4,119 (10.5) | 4,004 (10.7) | 10,978 (12.3) |
| Repeat with labour | 39 (0.4) | 27 (0.4) | 20 (0.3) | 767 (0.6) | 442 (0.6) | 749 (0.5) | 385 (0.8) | 201 (0.8) | 865 (0.7) | 428 (1.1) | 365 (1.0) | 821 (0.9) |
| Repeat with no labour | 504 (4.9) | 331 (5.1) | 300 (5.0) | 9,241 (6.97) | 5,304 (7.2) | 9,364 (6.6) | 4,980 (10.9) | 2,445 (9.3) | 10,999 (9.6) | 6,034 (15.4) | 5,390 (14.4) | 12,291 (13.8) |
| One previous C-section | 558 (5.4) | 357 (5.5) | 310 (5.2) | 9,734 (7.34) | 5,475 (7.5) | 9,484 (6.7) | 4,666 (10.2) | 2,209 (8.4) | 10,355 (9.0) | 5,018 (12.8) | 4,356 (11.6) | 10,397 (11.6) |
| ≥1 previous C-section | 106 (1.0) | 85 (1.3) | 77 (1.3) | 2,411 (1.8) | 1,570 (2.1) | 2,464 (1.7) | 1,699 (3.7) | 961 (3.7) | 3,312 (2.9) | 2,160 (5.5) | 2,069 (5.5) | 3,979 (4.4) |
| **Induced labour** | 1,853 (18.1) | 1,046 (16.1) | 1,241 (20.8) | 26,284 (19.8) | 12,733 (17.4) | 33,785 (23.8) | 9,797 (21.4) | 4,902 (18.6) | 29,672 (25.9) | 10,105 (25.8) | 9,268 (24.8) | 25,867 (29.0) |
| **PROM (≥12 hours)** | 542 (5.3) | 322 (4.9) | 327 (5.5) | 6,759 (5.1) | 3,125 (4.3) | 8,821 (6.2) | 2,066 (4.5) | 969 (3.7) | 6,511 (5.7) | 1,604 (4.1) | 1,599 (4.3) | 4,684 (5.2) |
| **Precipitous labour** | 338 (3.3) | 300 (4.6) | 146 (2.4) | 4,758 (3.6) | 3,065 (4.2) | 3,693 (2.6) | 1,556 (3.4) | 995 (3.8) | 2,947 (2.6) | 1,058 (2.7) | 1,116 (3.0) | 1,895 (2.1) |
| **Prolonged labour (≥20 hours)** | 210 (2.1) | 110 (1.7) | 138 (2.3) | 2,825 (2.1) | 1,234 (1.7) | 3,648 (2.6) | 876 (1.9) | 404 (1.5) | 2,954 (2.6) | 827 (2.1) | 641 (1.7) | 2,287 (2.6) |
| **Gestational hypertension** | 228 (2.2) | 107 (1.6) | 226 (3.8) | 3,320 (2.5) | 1,547 (2.1) | 7,333 (5.2) | 1,738 (3.8) | 808 (3.1) | 7,862 (6.9) | 3,004 (7.7) | 2,379 (6.4) | 10,062 (11.3) |
| **Gestational diabetes** | 285 (2.8) | 265 (4.1) | 130 (2.2) | 4,713 (3.6) | 4,437 (6.1) | 3,652 (2.6) | 4,049 (8.8) | 2,622 (10.0) | 5,820 (5.1) | 5,507 (14.1) | 5,920 (15.8) | 8,670 (9.7) |
| **Chorioamnionitis** | 205 (2.0) | 113 (1.7) | 138 (2.3) | 2,541 (1.9) | 1,107 (1.5) | 3,558 (2.5) | 763 (1.7) | 342 (1.3) | 2,508 (2.2) | 536 (1.4) | 476 (1.3) | 1,700 (1.9) |

(*Continued*)

**Table 2.** (Continued)

| Characteristic | Pre-pregnancy underweight N = 22,714 (3.1%) | | | Pre-pregnancy normal BMI N = 347,555 (48.1%) | | | Pre-pregnancy overweight N = 186,662 (25.8%) | | | Pre-pregnancy obese N = 165,908 (23.0%) | | |
|---|---|---|---|---|---|---|---|---|---|---|---|---|
| | Optimal weight gain | Low weight gain | Excess weight gain | Optimal weight gain | Low weight gain | Excess weight gain | Optimal weight gain | Low weight gain | Excess weight gain | Optimal weight gain | Low weight gain | Excess weight gain |
| **Previous infant death or preterm or SGA infant** | 148 (1.4) | 102 (1.6) | 84 (1.4) | 1,756 (1.3) | 1,133 (1.5) | 1,858 (1.3) | 789 (1.7) | 477 (1.8) | 1,702 (1.5) | 767 (2.0) | 788 (2.1) | 1,713 (1.9) |

Data given as N (%). Underweight defined as BMI < 18.5 kg/m$^2$, normal weight as BMI 18.5–24.9 kg/m$^2$, overweight as BMI 25.0–29.9 kg/m$^2$, and obese class I–III as BMI $\geq$ 30.0 kg/m$^2$.

C-section, cesarean section; GA, gestational age; PROM, premature rupture of membranes; SGA, small-for-gestational age; VBAC, vaginal birth after cesarean section.

odds of maternal death/SMM compared with those with optimal weight gain (AOR 1.28, 95% CI 1.00–1.63, $p$ = 0.049; AOR 1.20, 95% CI 1.12–1.28, $p$ < 0.001; and AOR 1.12, 95% CI 1.01–1.23, $p$ = 0.019; respectively; Table 3). All women with excess weight gain had elevated odds of sepsis (mainly puerperal sepsis). In women who were underweight and obese, excess weight gain was also associated with higher odds of respiratory morbidity; those with normal weight had increased odds of postpartum hemorrhage with transfusion and potentially life-saving medical interventions; overweight women had higher odds of ICU admission (S3 Table).

## Perinatal outcomes

In each pre-pregnancy BMI category, women with low weight gain had higher odds of the composite adverse perinatal outcome—perinatal death and severe neonatal morbidity—than women with optimal weight gain (Table 3). Among women with low weight gain, the odds of stillbirth was significantly elevated in all pre-pregnancy BMI categories, except for a non-statistically-significant increase in underweight women (S4 Table).

**Table 3. Adjusted odds ratios (AORs) for severe adverse birth outcomes by gestational weight and pre-pregnancy BMI—Singleton births, Washington State, 2004–2013 (AOR relative to optimal weight gain in each pre-pregnancy BMI category).**

| Outcome | Pre-pregnancy underweight | | Pre-pregnancy normal BMI | | Pre-pregnancy overweight | | Pre-pregnancy obese | |
|---|---|---|---|---|---|---|---|---|
| | Low weight gain | Excess weight gain | Low weight gain | Excess weight gain | Low weight gain | Excess weight gain | Low weight gain | Excess weight gain |
| Maternal death/severe maternal morbidity[a] | 0.96 (0.74–1.24) *0.737* | **1.28 (1.00–1.63)** *0.049* | **1.12 (1.04–1.21)** *0.004* | **1.20 (1.12–1.28)** *<0.001* | **1.17 (1.04–1.32)** *0.009* | 1.07 (0.98–1.18) *0.209* | 1.07 (0.95–1.02) *0.244* | **1.12 (1.01–1.23)** *0.019* |
| Perinatal death[b] | **2.03 (1.26–3.29)** *0.003* | 1.28 (0.75–2.20) *0.369* | **2.14 (1.86–2.46)** *<0.001* | **1.41 (1.24–1.61)** *<0.001* | **2.53 (2.09–3.07)** *<0.001* | 0.87 (0.73–1.04) *0.099* | **2.04 (1.72–2.42)** *<0.001* | **0.82 (0.69–0.97)** *0.032* |
| Perinatal death/severe neonatal morbidity[b] | **1.92 (1.50–2.46)** *<0.001* | **1.49 (1.15–1.95)** *0.003* | **1.41 (1.31–1.53)** *<0.001* | **1.38 (1.29–1.85)** *<0.001* | **1.65 (1.47–1.84)** *<0.001* | 0.99 (0.90–1.08) *0.808* | **1.55 (1.40–1.72)** *<0.001* | 1.05 (0.96–1.15) *0.203* |
| Combined maternal and/or perinatal adverse outcome[b] | **1.35 (1.13–1.62)** *<0.001* | **1.38 (1.15–1.66)** *<0.001* | **1.26 (1.19–1.33)** *<0.001* | **1.28 (1.22–1.34)** *<0.001* | **1.39 (1.27–1.51)** *<0.001* | 1.02 (0.95–1.09) *0.575* | **1.32 (1.22–1.42)** *<0.001* | **1.09 (1.01–1.16)** *0.017* |

Values given as AOR (95% CI) and $p$-value. Two-sided $p$-values were calculated using multivariable logistic regression Wald chi-squared test; statistically significant AORs given in bold.

[a]Adjusted for maternal age (<25 years, 25–34 years, or $\geq$35 years), maternal education (high school graduation or higher versus less than high school graduation), marital status (single, widowed, or separated versus married or cohabitating), race/ethnicity (Hispanic, African American, Native American, or other versus non-Hispanic white), parity (nulliparous or parity $\geq$ 4 versus parity 1–3), assisted conception (no versus yes), smoking during pregnancy (no versus yes), type of health insurance (Medicaid versus private, self-pay, or other), year of birth, and fetal sex (female versus male).

[b]Adjusted for congenital anomalies in addition to covariates above.

**Table 4. Adjusted odds ratios (AORs) for severe adverse outcomes by gestational weight gain and obesity class—Singleton births, Washington State, 2004–2013 (AOR relative to optimal weight gain in each obesity class category).**

| Outcome | Pre-pregnancy BMI | | | | | |
| --- | --- | --- | --- | --- | --- | --- |
| | Obesity class I | | Obesity class II | | Obesity class III | |
| | Low weight gain | Excess weight gain | Low weight gain | Excess weight gain | Low weight gain | Excess weight gain |
| Maternal death/severe maternal morbidity[a] | 1.05 (0.90–1.25) *0.493* | 1.07 (0.94–1.22) *0.298* | 1.10 (0.88–1.36) *0.397* | 1.20 (1.00–1.44) *0.05* | 1.03 (0.81–1.30) *0.816* | 1.23 (0.99–1.54) *0.059* |
| Perinatal death/severe neonatal morbidity[b] | **1.52 (1.30–1.76)** *<0.001* | 1.04 (0.91–1.18) *0.441* | **1.68 (1.39–2.02)** *<0.001* | 1.06 (0.89–1.27) *0.375* | **1.42 (1.15–1.76)** *0.002* | 1.16 (0.94–1.44) *0.180* |
| Combined maternal and perinatal severe adverse outcomes[b] | **1.29 (1.15–1.44)** *<0.001* | 1.05 (0.96–1.16) *0.244* | **1.40 (1.21–1.61)** *<0.001* | 1.12 (0.98–1.28) *0.081* | **1.24 (1.06–1.46)** *<0.001* | **1.21 (1.03–1.42)** *0.016* |

Values given as AOR (95% CI). Two-sided *p*-values were calculated using multivariable logistic regression Wald chi-squared test; statistically significant AORs given in bold.

[a]Adjusted for maternal age (<25 years, 25–34 years, or ≥35 years), maternal education (high school graduation or higher versus less than high school graduation), marital status (single, widowed, or separated versus married or cohabitating), race/ethnicity (Hispanic, African American, Native American, or other versus non-Hispanic white), parity (nulliparous or parity ≥ 4 versus parity 1–3), assisted conception (no versus yes), smoking during pregnancy (no versus yes), type of health insurance (Medicaid versus private, self-pay, or other), year of birth, and fetal sex (female versus male).
[b]Adjusted for congenital anomalies in addition to covariates above.

In underweight and normal weight women, excess weight gain was associated with higher odds of perinatal death/severe neonatal morbidity compared with those with optimal weight gain (Table 3). Among women with normal pre-pregnancy BMI, those who had excess gestational weight gain had increased odds of perinatal death/severe neonatal morbidity, while in overweight women, excess weight gain did not increase the odds of any severe adverse perinatal outcomes. Obese women with excess weight gain had lower odds of perinatal death than obese women with optimal weight gain (Table 3 and S4 Table).

## Secondary analyses including only obese women

Regardless of the pre-pregnancy obesity class, obese women with low or excess gestational weight gain did not have a significantly increased odds of the composite outcome maternal death/SMM (Table 4). An increased odds of ICU admission was observed among women who had pre-pregnancy class I obesity and low gestational weight gain (S6 Table). Excess gestational weight gain was significantly associated with increased odds of sepsis among women with class II obesity (S6 Table). Perinatal death/severe neonatal morbidity was elevated among women with low weight gain in each pre-pregnancy obesity class group (Table 4).

## Sensitivity analyses

Results restricted to women with term pregnancy included 672,141 women (93% of all women; S7 Table). Among women with normal pre-pregnancy BMI, both low and excess weight gain were associated with increased odds of maternal death/SMM compared with optimal weight gain. The increased risk of perinatal death/severe neonatal morbidity was no longer observed, except for obese women with low and excess weight gain (S7 Table).

## Discussion

In this study, we found that most women do not achieve optimal weight gain during pregnancy. Low weight gain was consistently associated with increased risk of severe adverse birth outcomes, and in particular with maternal death and perinatal death. On the other hand,

excess weight gain was associated with adverse maternal and perinatal outcomes only in women who had pre-pregnancy BMI in the normal range and below. Women who were overweight prior to pregnancy and gained excess weight during pregnancy did not have elevated risks of serious adverse birth outcomes compared with overweight women with optimal weight gain. Pre-pregnancy obesity and excess weight gain was associated with elevated risk of SMM.

Our results show that only about one-third of women achieved optimal weight gain, while almost half gained excess weight during pregnancy. The excess weight gain was more common in overweight and obese women (61% and 53%, respectively), which is consistent with the literature [29,30]. Previous studies underscored increases in SMM only among women with excess weight gain [14,16]; however, our results show an increase in maternal death/SMM also in women with low gestational weight gain. Prior studies were smaller and were often restricted to women with term pregnancy [14,16]. Our sensitivity analyses show that such restriction provides different results for perinatal outcomes, as women with low and excess weight gain are more likely to deliver preterm. Two large meta-analyses focusing on less severe adverse birth outcomes showed elevated rates of LGA and macrosomia in women with excess weight gain in all pre-pregnancy BMI categories, and elevated rates of SGA in women with low weight gain [2,31], which is consistent with our results. Low weight gain is known to be associated with preterm birth [4,5] and with infant death during the first year after birth [13,32]. The new findings in our study include the association between low weight gain and adverse perinatal outcomes, including stillbirth and neonatal death. Our results suggest that low weight gain during the second and third trimester can potentially serve as a marker for an increased risk of stillbirth. As such, weight gain monitoring may help to prevent stillbirth by increased prenatal surveillance, ultrasound checks, and early delivery in cases of worsened intrauterine growth restriction.

We did not observe any association between excess weight gain in overweight women and severe adverse maternal and perinatal outcomes, confirming the findings of Platner et al. [16] from a large study of pregnant women in New York City. This finding, together with the very high rate of excess weight gain in this group, suggests that the IOM/ACOG guidelines for optimal weight gain in these women may need a re-evaluation.

With respect to individual maternal morbidities, the association between excess weight gain and pregnancy complications requiring blood transfusion and ventilation is consistent with the literature [14,16]. In addition, we observed an increased risk of sepsis, especially puerperal sepsis, among women with excess weight gain regardless of pre-pregnancy BMI. This may indicate a potential link between excess weight gain and maternal immune response or placental function, which is consistent with a previously reported association between excess weight gain and chorioamnionitis [14]. Of note is also an elevated rate of gestational diabetes among women with low weight gain, consistently observed in all pre-pregnancy BMI groups. Such an association has been reported previously [14], suggesting that gestational diabetes may affect optimal nutritional intake in the second and third semesters, and potentially lead to adverse perinatal outcomes [33]. It is not entirely clear whether gestational diabetes control leads to low weight gain or whether another common cause leads to both gestational diabetes and low weight gain, and subsequently negatively impacts perinatal outcomes. While low and excess weight gain are potentially modifiable risk factors and may be causally related to the adverse outcomes observed, suboptimal weight gain can also result from underlying medical conditions and therefore serve as a "marker" for adverse pregnancy and fetal/infant health outcomes.

Our study has several strengths. We used a large population database with consistent data collection of important maternal and birth characteristics. The linkage between birth certificate and hospitalization data improved data accuracy (positive and negative predictive values

for most birth outcomes were greater than 80% and 98%, respectively) compared with similar non-linked databases [22].

This study has some limitations. First, information on pre-pregnancy BMI was self-reported. Previous studies have shown that self-reported weight measurements are usually within close range of the actual weight and are reliable [34,35]. However, if women intentionally reported lower pre-pregnancy weight than their actual weight, for instance due to social desirability, the proportion of those with excess weight gain would have been overestimated. This overestimation may have led to bias towards the null in our results pertaining to excess weight gain, especially among women with normal and higher pre-pregnancy BMI. Second, the IOM/ACOG recommendations for per-week weight gain are uniform for each gestational week preterm. We calculated optimal weight gain in women who delivered preterm by multiplying weekly recommended minimum (and maximum) weight gain by the number of weeks. However, gestational weight gain does not naturally occur in a linear fashion and tends to accelerate as pregnancy progresses during the second and third trimester. Thus, some women with early preterm deliveries may appear to have had lower than recommended weight gain, while their gain was in fact optimal [36]. Nevertheless, we aimed to follow the IOM/ACOG per-week recommendations in a pragmatic manner, the same way as they are used by maternity care providers. Third, we relied on ICD-9-CM codes to identify severe maternal conditions, which may be under-reported [37,38], resulting in potential underestimation of the associations with suboptimal weight gain. Similarly, potential omissions and coding errors in other covariates may have led to non-differential misclassification and may have biased the results towards the null [38]. Fourth, we did not have information on some behavioural factors, such as alcohol/drug use, that are associated with weight gain and adverse pregnancy outcomes. However, we did not aim to examine potential causes or contributors to suboptimal weight gain and their direct or indirect effects on adverse outcomes. We did adjust, however, for common demographic factors including marital status, race, education, and type of medical insurance. Last, our finding that more than half of women did not achieve optimal weight gain during pregnancy may be influenced by the ethnic/racial composition and other characteristics of the population of pregnant women in Washington State. And while similar proportions with suboptimal weight gain have been observed in the US and Europe, in Asian countries, for instance, the proportion of women with excess weight gain is much lower when standard BMI categories are applied [31].

This study adds to the body of literature that underscores the importance of pre-pregnancy BMI and weight gain during pregnancy with respect to severe adverse pregnancy outcomes. And while these 2 risk factors are modifiable, we still need effective strategies for women to achieve optimal BMI and weight gain during pregnancy. Pregnancy is an important opportunity to advocate for and support women in behavioural changes towards a healthy lifestyle.

To our knowledge, this is the first large study to assess the associations between serious adverse outcomes and suboptimal weight gain during pregnancy in both term and preterm gestations, making our results more generalizable and applicable to all women in the second and third trimester of pregnancy and their healthcare providers. Our findings are comprehensive with respect to all serious adverse outcomes (death and severe morbidity) that are relevant to mother, fetus, and infant. This facilitates counselling for pregnant women about severe adverse effects associated with weight gain during pregnancy, according to their pre-pregnancy BMI. Future research is needed to create algorithms that would identify optimal weight gain trajectories tailored to individual women, based on their pre-pregnancy BMI and other prognostic factors. In addition, future research is required to investigate the potential utility of low weight gain as an early sign of increased risk of stillbirth.

## Supporting information

**S1 Form. Washington State birth filing form sample.**
(DOCX)

**S1 STROBE Checklist. STROBE checklist of items that should be included in reports of observational studies.**
(DOCX)

**S1 Table. Severe maternal morbidity definitions.**
(DOCX)

**S2 Table. Rates of maternal death and severe maternal morbidity (rates per 10,000 births) by pre-pregnancy BMI and gestational weight gain.**
(DOCX)

**S3 Table. Adjusted odds ratios (AORs) for SMM components by gestational weight gain and pre-pregnancy BMI.**
(DOCX)

**S4 Table. Adjusted odds ratios (AORs) for stillbirth, neonatal death, and severe neonatal morbidity by gestational weight gain and pre-pregnancy BMI.**
(DOCX)

**S5 Table. Rates of maternal death and severe maternal morbidity by gestational weight gain and pre-pregnancy obesity class.**
(DOCX)

**S6 Table. Adjusted odds ratios (AORs) for the components of SMM by gestational weight gain and pre-pregnancy obesity class.**
(DOCX)

**S7 Table. Adjusted odds ratios (AORs) for severe adverse maternal and perinatal outcomes by gestational weight gain and pre-pregnancy BMI—Singleton births at term gestation ($\geq$37 weeks).**
(DOCX)

**S1 Text. Study analysis plan.**
(DOCX)

## Author Contributions

**Conceptualization:** U. Vivian Ukah, Hamideh Bayrampour, Yasser Sabr, Neda Razaz, Wee-Shian Chan, Kenneth I. Lim, Sarka Lisonkova.

**Data curation:** Sarka Lisonkova.

**Formal analysis:** U. Vivian Ukah.

**Funding acquisition:** Sarka Lisonkova.

**Methodology:** U. Vivian Ukah, Hamideh Bayrampour, Yasser Sabr, Neda Razaz, Wee-Shian Chan, Kenneth I. Lim, Sarka Lisonkova.

**Supervision:** Sarka Lisonkova.

**Validation:** Hamideh Bayrampour, Yasser Sabr, Neda Razaz, Wee-Shian Chan, Kenneth I. Lim.

**Writing – original draft:** U. Vivian Ukah.

**Writing – review & editing:** Hamideh Bayrampour, Yasser Sabr, Neda Razaz, Wee-Shian Chan, Kenneth I. Lim, Sarka Lisonkova.

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
