## [Decision Letter · Decision Letter 0]

30 Sep 2019

Dear Dr. Ukah,

Thank you very much for submitting your manuscript "Association between gestational weight gain and severe adverse birth outcomes: Does pre-pregnancy body mass index matter?" (PMEDICINE-D-19-02499) for consideration at PLOS Medicine. 

[LINK]

In light of these reviews, I am afraid that we will not be able to accept the manuscript for publication in the journal in its current form, but we would like to consider a revised version that addresses the reviewers' and editors' comments. Obviously we cannot make any decision about publication until we have seen the revised manuscript and your response, and we plan to seek re-review by one or more of the reviewers. 

We expect to receive your revised manuscript by Oct 21 2019 11:59PM. Please email us (plosmedicine@plos.org) if you have any questions or concerns.

We look forward to receiving your revised manuscript. 

Sincerely,

Clare Stone, PhD

Acting Chief Editor 

PLOS Medicine

plosmedicine.org

Title – please add a study design in title, and remove rhetorical question

Please comment on the limitation of the data and how old it is.

Please provide a summary of demographic details and limitations in abstract (limitations as the final sentence of the Methods and Findings section.)

Please provide p values where 95%Cis are given.

data – please be explicit about why access to data is prohibited (IRB or ethical restrictions, for example).Also we need the email or URL for (eligible) people to access data. Noting this cannot be an author. 

Ethics – please say why it was exempt. 

Did your study have a prospective protocol or analysis plan? Please state this (either way) early in the Methods section.

c) In either case, changes in the analysis—including those made in response to peer review comments—should be identified as such in the Methods section of the paper, with rationale.

at lines 55 and 345, "In this study ... we found that ... was ..." or similar

Please ensure that the study is reported according to the [STROBE] guideline, and include the completed [STROBE or other] checklist as Supporting Information. When completing the checklist, please use section and paragraph numbers, rather than page numbers. Please add the following statement, or similar, to the Methods: "This study is reported as per the Strengthening the Reporting of Observational Studies in Epidemiology (STROBE) guideline (S1 Checklist)."

References should be in square brackets instead of round throughout text. 

Line 397 – I wonder if “We did not aim to examine causal associations” should be removed. As this is an observational study, causality couldn’t be assessed anyway. Otherwise rephrase to outline that this is not possible in this study. 

Please use the "Vancouver" style for reference formatting, and see our website for other reference guidelines https://journals.plos.org/plosmedicine/s/submission-guidelines#loc-references

Please ensure that the study is reported according to the STROBE guideline, and include the completed STROBE checklist as Supporting Information. Please add the following statement, or similar, to the Methods: "This study is reported as per the Strengthening the Reporting of Observational Studies in Epidemiology (STROBE) guideline (S1 Checklist)."

Comments from the reviewers:

Reviewer #1: This is a well-conducted study on the association between gestational weight gain and severe adverse birth outcomes, analysed in different pre-pregnancy BMI categories. The study design, datasets, statistical methods and analyses, and presentation (tables and figures) and interpretations of results are mostly adequate and of a good standard. Only a couple of minor points needing attention:

1) Table 2 and 3 are massive tables with a lot of details which are not as important as other tables in terms of primary analyses. To be concise, suggest to remove both or one of the tables to the supplementary information.

2) Table 1 is a huge descriptive table but without any statistical test. I know it's already huge, but it would be helpful to add a column of p-values in each of the pre-pregnancy BMI categories to test the statistical significance of the baseline characteristics across the levels of weight gain, which will make it easy for the readers to read and follow the main message.

Reviewer #2: This large observational study shows adverse health impacts of inadequate or excess gestational weight gain, with a focus on severe morbidity and mortality. Linked data sets and sheer size of the analysis are strengths of this work. 

However, the limitations need also to be addressed. The findings are largely consistent with prior literature. The clear gaps proposed in the introduction are not strongly justified. The studies referenced in the introduction have previously addressed many of the key gaps in this field and the contribution of prior literature and the very specific nuances aims and hypotheses should be more clearly outlined. It is not clear what the specific initial hypotheses were and why?

It is not clear why ethics review was waivered and this should be further clarified, especially given the use of linked data bases.

A clear rationale for the composite outcomes should be provided. 

The observation on line 221 that those with GDM were more likely to have low GWG is expected as treatment of GDM reduces maternal weight gain. This presents an interesting subgroup to study relationships between GWG and outcomes and could perhaps be explored as a subgroup here.

The data presented here reflects the nuances of the specific ethnicities in the US and the unique health system in the US, both of which are limitations for generalisability. This is especially regarding the health system which requires around 20% GDP and has significant access and disparity issues, limiting generalisability and this should be acknowledged. Likewise the ethnic variation shown to vary in Asian subgroups specifically (eg Goldstein et al BMC Medicine), limits the broader generalisability for example in the majority of the international population who are of Asian ethnicity. 

Table 4 is difficult to interpret, as there seems to be a separate column of data superimposed on the last column. 

The focus on the inconsistent findings of rare outcomes across multiple subgroups as noted in the first paragraph on the discussion, are difficult to interpret and appear inconsistent. For example findings in normal and obese but not overweight women. These subgroup results appear inconsistent and not driven by plausible hypotheses, an area that needs to be more thoroughly explored. 

Broader existing literature, evidence synthesis and meta-analyses need to be considered in the discussion. For example the discussion in lines 373-377 should be revised to extend to evidence that does not support these findings including meta analyses and the potential reasons why. Specifically the differences from JAMA. 2019;321(17):1702-1715. doi:10.1001/jama.2019.3820 and JAMA. 2017;317(21):2207-2225. doi:10.1001/jama.2017.3635 as large scale intergrated evidence synthesis studies and the findings here should be discussed in this context. 

Reviewer #3: The purpose of this study was to investigate the association between suboptimal and excessive weight gain and the effects on adverse birth outcomes in women with a range of pre-pregnancy BMI. This was a population-based study of a singleton hospital births in the state of Washington between 2004 and 2013. The primary composite outcome included: 1) maternal death or severe maternal morbidity, and 2) perinatal death and/or severe neonatal morbidity. Overall 722,839 women with information on pre-pregnancy BMI were included in the analysis. The distribution of BMI weight classes, according to WHO criteria, was consistent with what has been previously reported. Low gestational weight gain was associated with higher rates of maternal death and severe maternal morbidity in women with normal pre-pregnancy BMI and overweight women. Similarly, excessive gestational weight gain was associated with an increased rate of death and severe maternal morbidity among women with normal pre-pregnancy BMI and obese women. Low gestational weight gain was associated with perinatal death and severe neonatal morbidity regardless of pre-pregnancy BMI, including obese classes 1, 2, and 3. The authors conclude that most women do not achieve optimal weight during pregnancy. Both low and excessive gestational weight gain were associated with severe adverse birth outcomes regardless of pre-pregnancy BMI except for overweight women with excessive gestational weight gain. 

Specific Comments:

1. This is a well-written manuscript with detailed tables, which because of the large number of cases could easily serve as a reference manuscript. The inclusion of both pre-term and term pregnancies, along with detailed information about class I, II, and III obesity, are important components of this manuscript.

2. As noted by the authors in the introduction, there was a lack of specific recommendations relating to gestational weight gain in the IOM report in class I, II and III obese women because of the lack of data in 2008. The information in this manuscript helps to improve that knowledge gap.

3. Gestational age at delivery was based on ultrasound dating. One assumes that the ultrasound dating in the vast majority of subjects was done in early pregnancy. It would be helpful if the authors would provide information on gestational age at the time of the ultrasound. 

4. Pregnancy weight gain was defined as a woman's weight measured at childbirth subtracted from her pre-pregnancy self-reported weight. Many investigators use weight at the last ante-natal visit as the end point of weight gain during pregnancy. In the state of Washington, was maternal weight actually measured in the labor and delivery unit?

5. To account for gestational age at delivery among women who delivered pre-term, weight pre-term was calculated using the IOM criteria. These criteria provide a range of weight gain per week. Which part of the range was used in the analysis and was it used consistently among the various BMI groups?

6. Very detailed information is provided in both the tables in the manuscript and in the supplementary tables. Is all of this this information available in the BERD and CHARS data bases or were additional means used to attain the granularity of detail presented in these tables?

7. The percentage of women in each pre-pregnancy BMI weight category is pretty much consistent as what has been reported in the literature, including data from the CDC.

8. On the bottom of page 11, women with low weight gain had higher rates of spontaneous vaginal delivery and lower rates of prolonged labor. The rates of gestational diabetes were higher in these women, especially for those who were underweight and those with a normal BMI prior to pregnancy. These data are very interesting since many previous studies have related the increased risk of gestational diabetes on excessive gestational weight gain and not low gestational weight gain. 

9. The data presented in table 3 at the bottom of page 18 are important as noting that low gestational weight gain had a significantly higher rate of maternal death as compared with women with optimal weight gain (7.97 vs. 2.63 per 100,000 p=0.027). Among normal weight and overweight women, those with low gestational weight gain had higher adjusted odds of maternal death and severe morbidity compared with optimal gestational weight gain. These data are important to note given that many individuals recommended gestational weight gain less than IOM guidelines to improve some perinatal outcomes.

10. Similarly in Table 4, it is important to note that the adjusted odds ratio for severe adverse birth outcomes is significantly higher in obese women with low weight gain for perinatal death and severe neonatal morbidity, as well as the combined maternal and perinatal adverse outcomes. 

11. On page 26, lines 373-377, the authors state that they did not observe any association between excessive weight gain in overweight women in severe adverse maternal and perinatal outcomes. Although these women had a very high rate of excessive weight gain, the authors suggest that the IOM guidelines for optimal weight gain in overweight women may need to be re-evaluated. How would the authors recommend changing the IOM guidelines for overweight women based on their findings?

12. In multiple places in the manuscript, for example page 3 line 84 the authors refer to the IOM/AJOG guidelines. AJOG should be ABOG.

[LINK]

---

## [Decision Letter · Decision Letter 1]

19 Nov 2019

Dear Dr. Ukah,

Thank you very much for re-submitting your manuscript "Association between gestational weight gain and severe adverse birth outcomes: Does pre-pregnancy body mass index matter?" (PMEDICINE-D-19-02499R1) for review by PLOS Medicine.

I have discussed the paper with my colleagues and the academic editor and it was also seen again by xxx reviewers. I am pleased to say that provided the remaining editorial and production issues are dealt with we are planning to accept the paper for publication in the journal.

[LINK]

We look forward to receiving the revised manuscript by Nov 26 2019 11:59PM. 

Sincerely,

Louise Gaynor-Brook, MBBS PhD

Associate Editor 

PLOS Medicine

plosmedicine.org

Requests from Editors:

General comments

Please add reference brackets at the end of each sentence, before the full stop and and after a space following the last word.

Please remove ‘Appendix’ in references to Supplementary Information files

Data Availability: The data link provided takes us to a paid for data access site, costing $50 per file, per year. Access to data via a paid service does not comply with our data policy. Please provide a non-payment website where data can be accessed / where authors can apply for access. Please note that a study author cannot be the contact person for the data. 

Title – Please include setting and dates of study. We suggest revising the title to “Association between gestational weight gain and severe adverse birth outcomes in Washington, USA: A population-based, retrospective cohort study, 2004-2013”

Abstract

Please begin the Methods and Findings with ‘We conducted’ or similar.

Lines 42-45: Please consider breaking this sentence up into 3 shorter sentences. 

Lines 50-52: Please clarify what is meant by ‘a composite maternal death’ in sentence beginning ‘In addition, low weight gain...’. 

Author Summary 

Line 71 - please add (IOM) after Institute of Medicine 

Introduction

Line 122 - please clarify what is meant by ‘delivery hospitalization’

Methods 

Line 142 - Please revise to ‘...guideline (S1 Checklist).’ Please rename the prospective analysis plan file as ‘S1 Text’ and include a reference to this your Methods.

Results 

Table 2 - please provide definitions for all abbreviated terms used, in the table legend.

Tables 3 & 4 - When a p value is given, please specify the statistical test used to determine it. 

Discussion 

Line 407 - please reformat reference in superscript to PLOS house style (please see below).

Please add a short section after discussing the strengths/limitations to discuss the implications of the study and next steps for research, clinical practice, and/or public policy

Please remove the subtitle of ‘Conclusion’

Line 482 - please correct to ‘trimester’

Figure 1 – Please revise figure to use solid blocks of colour (rather than patterned), and ideally avoid using green and red together. 

References

Please make sure references are appropriately capitalised and formatted e.g. ref 11 ‘BMC pregnancy and childbirth’ to ‘BMC Pregnancy and childbirth’

Please use the "Vancouver" style for reference formatting, and see our website for other reference guidelines https://journals.plos.org/plosmedicine/s/submission-guidelines#loc-references

Supplementary Files 

S1 STROBE Checklist - When completing the checklist, please use section and paragraph numbers, rather than line numbers.

Tables S3, S4, S6, S7 - please provide definitions for all abbreviated terms used, in the table legend

Tables S4, S5, S8 - When a p value is given, please specify the statistical test used to determine it. 

Comments from Reviewers:

Reviewer #1: I am satisfied with the response and the revision. No further issues needing attention.

Reviewer #3: The authors have responded to all my queries and I have no further comments of questions.

[LINK]

---

## [Editor Report · Decision Letter 2]

3 Dec 2019

Dear Dr Ukah, 

On behalf of my colleagues and the academic editor, Dr. Louise Gaynor-Brook, I am delighted to inform you that your manuscript entitled "Association between gestational weight gain and severe adverse birth outcomes in Washington State, USA: A population-based retrospective cohort study, 2004-2013" (PMEDICINE-D-19-02499R2) has been accepted for publication in PLOS Medicine. 

PRODUCTION PROCESS

PRESS

PROFILE INFORMATION

Thank you again for submitting the manuscript to PLOS Medicine. We look forward to publishing it. 

Best wishes, 

Louise Gaynor-Brook, MBBS PhD

Associate Editor 

PLOS Medicine

plosmedicine.org